# Precise atom manipulation through deep reinforcement learning

I-Ju Chen [1] ✉, Markus Aapro[1], Abraham Kipnis [1], Alexander Ilin[2], Peter Liljeroth [1] ✉ & Adam S. Foster [1,3] ✉

Atomic-scale manipulation in scanning tunneling microscopy has enabled the creation of quantum states of matter based on artificial structures and extreme miniaturization of computational circuitry based on individual atoms. The ability to autonomously arrange atomic structures with precision will enable the scaling up of nanoscale fabrication and expand the range of artificial structures hosting exotic quantum states. However, the a priori unknown manipulation parameters, the possibility of spontaneous tip apex changes, and the difficulty of modeling tip-atom interactions make it challenging to select manipulation parameters that can achieve atomic precision throughout extended operations. Here we use deep reinforcement learning (DRL) to control the real-world atom manipulation process. Several state-of-the-art reinforcement learning (RL) techniques are used jointly to boost data efficiency. The DRL agent learns to manipulate Ag adatoms on Ag(111) surfaces with optimal precision and is integrated with path planning algorithms to complete an autonomous atomic assembly system. The results demonstrate that state-of-the-art DRL can offer effective solutions to real-world challenges in nanofabrication and powerful approaches to increasingly complex scientific experiments at the atomic scale.

Since its first demonstration in the 1990s[1], atom manipulation using a scanning tunneling microscope (STM) is the only experimental technique capable of realizing atomically precise structures for research on exotic quantum states in artificial lattices and atomic-scale miniaturization of computational devices. Artificial structures on metal surfaces allow tuning electronic and spin interactions to fabricate designer quantum states of matter[2–8]. Recently, atom manipulation has been extended to platforms including superconductors[9,10], 2D materials[11–13], semiconductors[14,15], and topological insulators[16] to create topological and many-body effects not found in naturally occurring materials. In addition, atom manipulation is used to build and operate computational devices scaled to the limit of individual atoms, including quantum and classical logic gates[17–20], memory[21,22], and Boltzmann machines[23].

Arranging adatoms with atomic precision requires tuning tip-adatom interactions to overcome energetic barriers for vertical or lateral adsorbate motion. These interactions are carefully controlled via the tip position, bias, and tunneling conductance set in the manipulation process[24–26]. These values are not known a priori and must be established separately for each new adatom/surface and tip apex combination. When the manipulation parameters are not chosen correctly, the adatom movement may not be precisely controlled, the tip can crash unexpectedly into the substrate, and neighboring adatoms can be rearranged unintentionally. In addition, fixed manipulation parameters may become inefficient following spontaneous tip apex structure changes. In such events, human experts generally need to search for a new set of manipulation parameters and/or reshape the tip apex.

In recent years, DRL has emerged as a paradigmatic method for solving nonlinear stochastic control problems. In DRL, as opposed to standard RL, a decision-making agent based on deep neural networks learns through trial and error to accomplish a task in dynamic

[1]Department of Applied Physics, Aalto University, Espoo, Finland. [2]Department of Computer Science, Aalto University, Espoo, Finland. [3]Nano Life Science Institute (WPI-NanoLSI), Kanazawa University, Kanazawa 920-1192, Japan. ✉e-mail: i-ju.chen@aalto.fi; peter.liljeroth@aalto.fi; adam.foster@aalto.fi

environments[27]. Besides achieving super-human performances in games[28,29] and simulated environments[30–32], state-of-the-art DRL algorithms' improved data efficiency and stability also opens up possibilities for real-world adoptions in automation[33–36]. In scanning probe microscopy, machine learning approaches have been integrated to address a wide variety of issues[37,38] and DRL with discrete action spaces has been adopted to automate tip preparation[39] and vertical manipulation of molecules[40].

In this work, we show that a state-of-the-art DRL algorithm combined with replay memory techniques can efficiently learn to manipulate atoms with atomic precision. The DRL agent, trained only on real-world atom manipulation data, can place atoms with optimal precision over 100 episodes after ~2000 training episodes. Additionally, the agent is more robust against tip apex changes than a baseline algorithm with fixed manipulation parameters. When combined with a path-planning algorithm, the trained DRL agent forms a fully autonomous atomic assembly algorithm which we use to construct a 42 atom artificial lattice with atomic precision. We expect our method to be applicable to surface/adsorbate combinations where stable manipulation parameters are not yet known.

## Results and discussion
### DRL implementation
We first formulate the atom manipulation control problem as a RL problem to solve it with DRL methods (Fig. 1a). RL problems are usually formalized as Markov decision processes where a decision-making agent interacts sequentially with its environment and is given goal-defining rewards. The Markov decision processes can be broken into episodes, with each episode starting from an initial state $s_0$ and terminating when the agent accomplishes the goal or when the maximum episode length is reached. Here the goal of the DRL agent is to move an adatom to a target position as precisely and efficiently as possible. In each episode, a new random target position 0.288 (one lattice constant $a$) – 2.000 nm away from the starting adatom position is given, and the agent can perform up to $N$ manipulations to accomplish the task. Here the episode length is set to an intermediate value $N = 5$ that allows the agent to attempt different ways to accomplish the goal without it being stuck in overly challenging episodes. The state $s_t$ at each discrete time step $t$ contains the relevant information of the environment. Here $s_t$ is a four-dimensional vector consisting of the $XY$-coordinates of the target position $\mathbf{x}_{target}$ and the current adatom position $\mathbf{x}_{adatom}$ extracted from STM images (Fig. 1(c)). Based on $s_t$, the agent selects an action $a_t \sim \pi(s_t)$ with its current policy $\pi$. Here $a_t$ is a six-dimensional vector comprised of the bias $V = 5$–15 mV (predefined range), tip-substrate tunneling conductance $G = 3$–6 μA/V, and the $XY$-coordinates of the start $\mathbf{x}_{tip,start}$ and end positions $\mathbf{x}_{tip,end}$ of the tip during the manipulation. Upon executing the action in the STM, a method combining a convolutional neural network and an empirical formula is used to classify whether the adatom has likely moved from the tunneling current measured during manipulation (see Methods section). If the method determines the adatom has likely moved, a scan is taken to update the adatom position to form the new state $s_{t+1}$. Otherwise, the scan is often skipped to save time and the state is considered unchanged $s_{t+1} = s_t$. The agent then receives a reward $r_t(s_t, a_t, s_{t+1})$. The reward signal defines the goal of the DRL problem. It is arguably the most important design factor, as the agent's objective is to maximize its total expected future rewards. The experience at each $t$ is

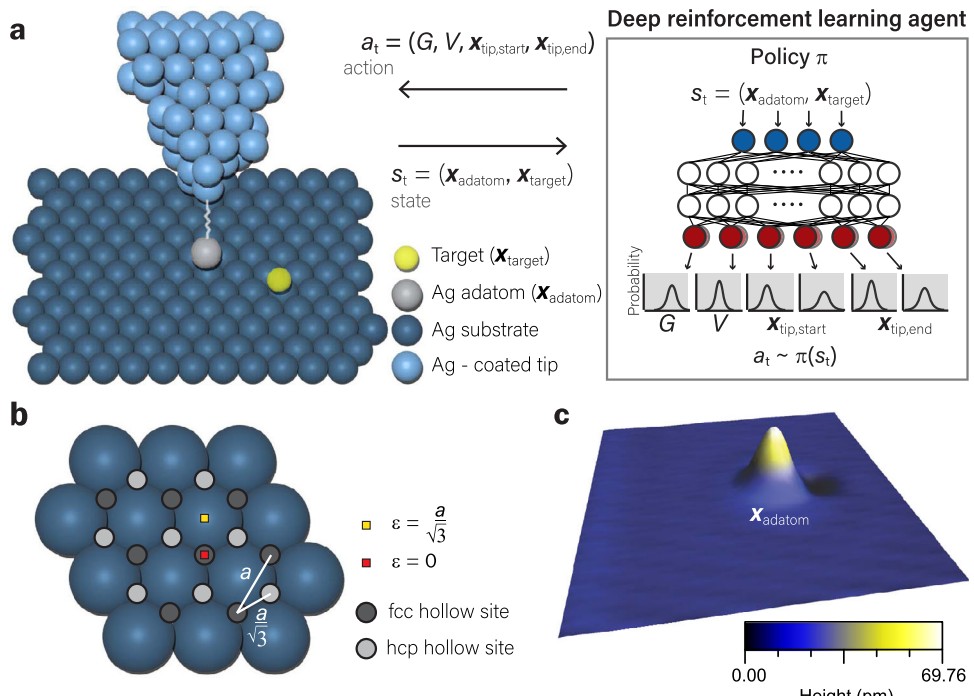

**Fig. 1 | Atom manipulation with a DRL agent. a** The DRL agent learns to manipulate atoms precisely and efficiently through interacting with the STM environment. At each $t$, an action command $a_t \sim \pi(s_t)$ is sampled from the DRL agent's current policy $\pi$ based on the current state $s_t$. The policy $\pi$ is modeled as a multivariate Gaussian distribution with mean and covariance given by the policy neural network. The action $a_t$ includes the conductance $G$, bias $V$, and the two-dimensional tip position at the start (end) of the manipulation $\mathbf{x}_{tip,start}$ ($\mathbf{x}_{tip,end}$), which are used to move the STM tip to try to move the adatom to the target position. **b** The atom manipulation goal is to bring the adatom as close to the target position as possible. For Ag on Ag(111) surfaces, the fcc (face-centered cubic) and hcp (hexagonal close-packed) hollow sites are the most energetically favorable adsorption sites[46,47]. From the geometry of the adsorption sites, the error $\varepsilon$ is limited to ranges from 0 nm to $\frac{a}{\sqrt{3}}$ depending on the target position. Therefore, the episode is considered successful and terminates if the $\varepsilon$ is lower than $\frac{a}{\sqrt{3}}$. **c** STM image of an Ag adatom on Ag substrate. Bias voltage 1 V, current setpoint 500 pA.

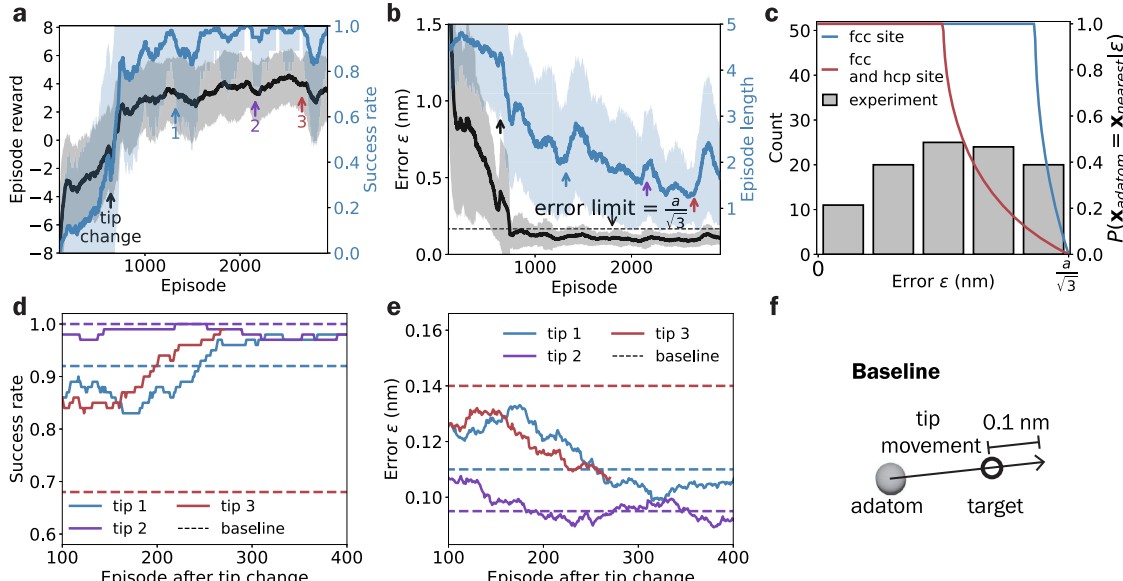

**Fig. 2 | DRL training results. a**, **b** The rolling mean (solid lines) and standard deviation (shaded areas) of episode reward, success rate, error, and episode length over 100 episodes showcase the training progress. The arrows indicate significant tip changes which occurred when the tip crashed deeply into the substrate and the tip apex needed to be reshaped to perform manipulation with the baseline parameters (see Methods) and the changes can be observed in the scan (see Supplementary Information). **c** The probability an atom is placed at the nearest adsorption site to the target at a given error $P(\mathbf{x}_{adatom} = \mathbf{x}_{nearest}|\varepsilon)$ is calculated considering either only fcc sites or both fcc and hcp sites (see Methods section). With the error distribution of the 100 consecutive successful training episodes, we estimate the atoms are placed at the nearest site ~93% (only fcc sites) and ~61% (both fcc and hcp sites) of the time. **d**, **e** The DRL agent, which is continually trained, and the baseline are compared under three tip conditions that resulted from the tip changes indicated in **a**, **b**. The baseline uses bias $V = 10$ mV, conductance $G = 6$ μA/V, and tip movements illustrated in **f**. Under the three tip conditions, the baseline manipulation parameters lead to varying performances. In contrast, DRL always converges to near-optimal performances after sufficient continued training. **f** In the baseline manipulation parameter, the tip moves from the adatom position to the target position extended by 0.1 nm.

stored in the replay memory buffer as a tuple $(s_t, a_t, r_t, s_{t+1})$ and used for training the DRL algorithm.

In this study, we use a widely adopted approach for assembling atom arrangements - lateral manipulation of adatoms on (111) metal surfaces. A silver-coated PtIr-tip is used to manipulate Ag adatoms on an Ag(111) surface at ~5 K temperature. The adatoms are deposited on the surface by crashing the tip into the substrate in a controlled manner (see Methods section). To assess the versatility of our method, the DRL agent is also successfully trained to manipulate Co adatoms on a Ag(111) surface (see Methods section).

Due to difficulties in resolving the lattice of the close-packed metal (111) surface in STM topographs[41], target positions are sampled from a uniform distribution regardless of the underlying Ag(111) lattice orientation. As a result, the optimal atom manipulation error $\varepsilon$, defined as the distance between the adatom and the target positions $\varepsilon := \|\mathbf{x}_{adatom} - \mathbf{x}_{target}\|$, is limited from 0 nm to $\frac{a}{\sqrt{3}} = 0.166$ nm, as shown in Fig. 1b and Methods, where $a = 0.288$ nm is the lattice constant on the Ag(111) surface. Therefore, in the DRL problem, the manipulation is considered successful and the episode terminates if $\varepsilon$ is smaller than $\frac{a}{\sqrt{3}}$. The reward is defined as

$$r_t(s_t, s_{t+1}) = \frac{-(\varepsilon_{t+1} - \varepsilon_t)}{a} + \begin{cases} -1 & \text{if } \varepsilon_{t+1} \geq \frac{a}{\sqrt{3}} \\ 1 & \text{if } \varepsilon_{t+1} < \frac{a}{\sqrt{3}} \end{cases}, \quad (1)$$

where the agent receives a reward +1 for a successful manipulation and −1 otherwise, and a potential-based reward shaping term[42] $\frac{-(\varepsilon_{t+1}-\varepsilon_t)}{a}$ that increases reward signals and guides the training process without misleading the agent into learning sub-optimal policies.

Here, we implement the soft actor-critic (SAC) algorithm[43], a model-free and off-policy RL algorithm for continuous state and action spaces. The algorithm aims to maximize the expected reward as well as the entropy of the policy. The state-action value function $Q$ (modeled with the critic network) is augmented with an entropy term. Therefore,

the policy $\pi$ (also referred to as the actor) is trained to succeed at the task while acting as randomly as possible. The agent is encouraged to take different actions that are similarly attractive with regard to expected reward. These designs make the SAC algorithm robust and sample-efficient. Here the policy $\pi$ and $Q$-functions are represented by multilayer perceptrons with parameters described in Methods. The algorithm trains the neural networks using stochastic gradient descent, in which the gradient is computed using experiences sampled from the replay buffer and extra fictitious experiences based on Hindsight Experience Replay (HER)[44]. HER further improves data efficiency by allowing the agent to learn from experiences in which the achieved goal differs from the intended goal. We also implement the Emphasizing Recent Experience sampling technique[45] to sample recent experience more frequently without neglecting past experience, which helps the agent adapt more efficiently when the environment changes.

## Agent training and performance

The agent's performance improves along the training process as reflected in the reward, error, success rate, and episode length, as shown in Fig. 2a, b. The agent minimizes manipulation error and achieves 100 % success rate over 100 episodes after ~2000 training episodes or equivalently 6000 manipulations, which is comparable to the amount of manipulations carried out in previous large-scale atom-assembly experiments[21,25]. In addition, the agent continues to learn to manipulate the adatom efficiently with more training, as shown by the decreasing mean episode length. Major tip changes (marked by arrows in Fig. 2a, b) lead to clear yet limited deterioration in the agent's performance, which recovers within a few hundreds more training episodes.

The training is ended when the DRL agent reaches near-optimal performance after each of the several tip changes. In the agent's best performance, it achieves a 100% mean success rate and 0.089 nm

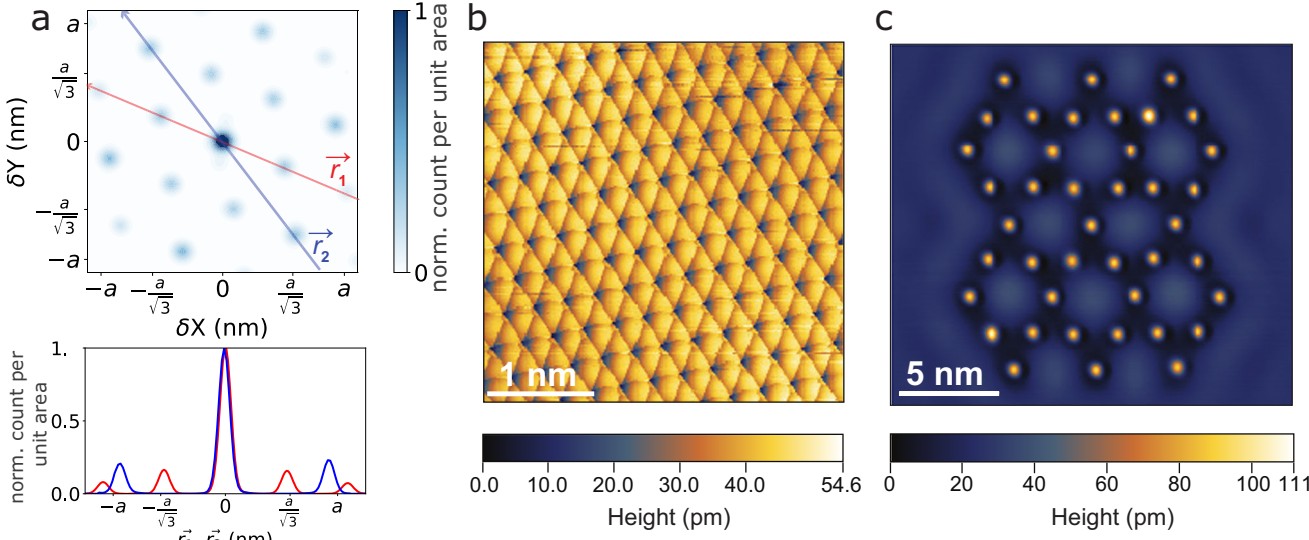

**Fig. 3 | Atom manipulation statistics and autonomous construction of an artificial lattice. a** Top: Adatom movement distribution following manipulations visualized in a Gaussian kernel density estimation plot. Adatoms are shown to reside both on fcc and hcp hollow sites. Line-cuts in two directions $\vec{r}_1$ and $\vec{r}_2$ (indicated by the blue and red arrows) are shown in the bottom figure. **b** Atomically resolved point contact scan obtained by manipulating an Ag atom. Bias voltage 2 mV, current 74.5 nA. The lattice orientation is in good agreement with **a**. **c** Together with the assignment and path-planning algorithms, the trained DRL agent is used to construct an artificial 42-atom kagome lattice with atomic precision. Bias voltage 100 mV, current setpoint 500 pA.

mean error over 100 episodes, significantly lower than one lattice constant (0.288 nm), and the error distribution is shown in Fig. 2c. Even though we cannot determine if the adatoms are placed in the nearest adsorption sites to the target without knowing the exact site positions, we can perform probabilistic estimations based on the geometry of the sites. For a given manipulation error $\varepsilon$, we can numerically compute the probability $P(\mathbf{x}_{adatom} = \mathbf{x}_{nearest}|\varepsilon)$ that an adatom is placed at the nearest site to the target for two cases: assuming that only fcc sites are reachable (the blue curve in Fig. 2c) and assuming that fcc and hcp sites are equally reachable (the red curve in Fig. 2c) (see Methods section). Then, using the obtained distribution $p(\varepsilon)$ of the manipulation errors (the gray histogram in Fig. 2c), we can estimate the probability that an adatom is placed at the nearest site

$$p(\mathbf{x}_{adatom} = \mathbf{x}_{nearest}) = \int p(\varepsilon)P(\mathbf{x}_{adatom} = \mathbf{x}_{nearest}|\varepsilon)d\varepsilon \quad (2)$$

to be between 61% (if both fcc and hcp sites are reachable) and 93% (if only fcc sites are reachable).

### Baseline performance comparison

Next, we compare the performance of the trained DRL algorithm with a set of manually tuned baseline manipulation parameters: bias $V = 10$ mV, conductance $G = 6$ μA/V, and tip movements shown in Fig. 2f under three different tip conditions (Fig. 2d, e). While the baseline achieves optimal performance under tip condition 2 (100% success rate over 100 episodes), the performances are significantly lower under the other two tip conditions, which have 92% and 68% success rates, respectively. In contrast, the DRL agent maintains relatively good performances within the first 100 episodes of continued training and eventually reaches success rates >95% after more training under the new tip conditions. The results show that, with continued training, the DRL algorithm is more robust and adaptable against tip changes than fixed manipulation parameters.

### Adsorption site statistics

The data collected during training also yields statistical insight into the adatom adsorption process and lattice orientation without atomically

resolved imaging. For metal adatoms on close-packed metal (111) surfaces, the fcc and hcp hollow sites are generally the most energetically favorable adsorption sites[46–48]. For Ag adatoms on the Ag(111) surface, the energy of fcc sites is found to be < 10 meV lower than hcp sites in theory[46] and STM manipulation experiments[47]. Here the distribution of manipulation-induced adatom movements from the training data shows that Ag adatoms can occupy both fcc and hcp sites, evidenced by the six peaks $\sim \frac{a}{\sqrt{3}} = 0.166$ nm from the origin (Fig. 3a). We also note that the adsorption energy landscape can be modulated by neighboring atoms and long-range interactions[49]. The lattice orientation revealed by the atom movements is in good agreement with the atomically resolved point contact scan in Fig. 3b.

### Artificial lattice construction

Finally, the trained DRL agent is used to create an artificial kagome lattice[50] with 42 adatoms shown in Fig. 3c. The Hungarian algorithm[51] and the rapidly-exploring random tree (RRT) search algorithm[52] break down the construction into single-adatom manipulation tasks with manipulation distance <2 nm, which the DRL agent is trained to handle. The Hungarian algorithm assigns adatoms to their final positions to minimize the total required movement. The RRT algorithm plans the paths between the start and final positions of the adatom while avoiding collisions between adatoms – note that it is possible that the structure in Fig. 3c contains 1 or 2 dimers, but these were likely formed before the manipulation started as the agent avoids atomic collisions. Combining these path planning algorithms with the DRL agent results in a complete software toolkit for robust, autonomous assembly of artificial structures with atomic precision.

The success in training a DRL model to manipulate matter with atomic precision proves that DRL can be used to tackle problems at the atomic level, where challenges arise due to mesoscopic and quantum effects. Our method can serve as a robust and efficient technique to automate the creation of artificial structures as well as the assembly and operation of atomic-scale computational devices. Furthermore, DRL by design learns directly from its interaction with the environment without needing supervision or a model of the environment, making it a promising approach to discover stable manipulation parameters that are not straightforward to human experts in novel systems.

In conclusion, we demonstrate that by combining several state-of-the-art RL algorithms and thoughtfully formalizing atom manipulation into the RL framework, the DRL algorithm can be trained to manipulate adatoms with atomic precision with excellent data efficiency. The DRL algorithm is also shown to be more adaptive against tip changes than fixed manipulation parameters, thanks to its capability to continuously learn from new experiences. We believe this study is a milestone in adopting artificial intelligence to solve automation problems in nanofabrication.

## Methods

### Experimental preparation

The Ag(111) crystal (MaTecK GmbH) is cleaned by several cycles of Ne sputtering (voltage 1 kV, pressure $5 \times 10^{-5}$ mbar) and annealing in UHV conditions ($p < 10^{-9}$ mbar). Atom manipulation is performed at ~5 K temperature in a Createc LT-STM/AFM system equipped with Createc DSP electronics and Createc STM/AFM control software (version 4.4). Individual Ag adatoms are deposited from the tip by gently indenting the apex to the surface[53]. For the baseline data and before training, we verify adatoms can be manipulated in the up, down, left and right directions with $V = 10$ mV and $G = 6$ μA/V following significant tip changes, and reshape the tip until stable manipulation is achieved. Gwyddion[54] and WSxM[55] software were used to visualize the scan data.

### Manipulating Co atoms on Ag(111) with deep reinforcement learning

In addition to Ag adatoms, DRL agents are also trained to manipulate Co adatoms on Ag(111). The Co atoms are deposited directly into the STM at 5 K from a thoroughly degassed Co wire (purity > 99.99%) wrapped around a W filament. Two separate DRL agents are trained to manipulate Co adatoms precisely and efficiently in two distinct parameter regimes: the standard close proximity range[56] with the same bias and tunneling conductance range as Ag (bias = 5–15 mV, tunneling conductance = 3–6 μA/V) shown in Suppl. Fig. 1 and a high-bias range[57] (bias = 1.5–3 V, tunneling conductance = 8–24 nA/V) shown in Suppl. Fig. 2. In the high-bias regime, a significantly lower tunneling conductance is sufficient to manipulate Co atoms due to a different manipulation mechanism. In addition, a high bias (~V) combined with a higher tunneling conductance (~μA/V) might lead to tip and substrate damage.

### Atom movement classification

STM scans following the manipulations constitute the most time-consuming part of the DRL training process. In order to reduce STM scan frequency, we developed an algorithm to classify whether the atom has likely moved based on the tunneling current traces obtained during manipulations. Tunneling current traces during manipulations contain detailed information about the distances and directions of atom movements with respect to the underlying lattice[25] as shown in Suppl. Fig. 3. Here we join a one-dimensional convolutional neural network (CNN) classifier and an empirical formula to evaluate whether atoms have likely moved during manipulations and if further STM scans should be taken to update their new positions. Due to the algorithm, STM scans are only taken after ~90% of the manipulations in the training shown in Fig. 2a, b.

### CNN classifier

The current traces are standardized and repeated/truncated to match the CNN input dimension = 2048. The CNN classifier has two convolutional layers with kernel size = 64 and stride = 2, a max pool layer with kernel size = 4 and stride = 2 and a dropout layer with a probability = 0.1 after each of them, followed by a fully connected layer with a sigmoid activation function. The CNN classifier is trained with the Adam optimizer with learning rate = $10^{-3}$ and batch size = 64. The CNN classifier is first trained on ~10,000 current

traces from a previous experiment. It reaches ~80% accuracy, true positive rate, and true negative rate on the test data. The CNN classifier is continuously trained with new current traces during DRL training.

### Empirical formula for atom movement prediction

We establish the empirical formula based on the observation that current traces often exhibit spikes due to atom movements, as shown in Suppl. Fig. 3. The empirical formula classifies atom movements as

$$\text{atom movement} = \begin{cases} \text{True} & \text{if } \frac{\partial I(\tau)}{\partial \tau} \geq c \cdot \sigma(I(\tau)) \\ \text{False} & \text{otherwise} \end{cases} \quad (3)$$

where $I(\tau)$ is the current trace as function of manipulation step $\tau$, $c$ is a tuning parameter set to 2–5 and $\sigma$ is the standard deviation.

In the DRL training, a STM scan is performed
- when the CNN prediction is positive;
- when the empirical formula prediction is positive;
- at random with probability ~20–40%; and
- when an episode terminates.

### Probability of atom occupying the nearest site as a function of $\varepsilon$

By analyzing the adsorption site geometry and integrating over possible target positions as shown in Suppl. Fig. 4, we compute the probability an atom is placed at the nearest site to the target at a given error $P(\mathbf{x}_{\text{adatom}} = \mathbf{x}_{\text{nearest}} | \varepsilon)$.

When only fcc sites are considered, we can observe the probability follows

$$P_{\text{fcc}}(\mathbf{x}_{\text{adatom}} = \mathbf{x}_{\text{nearest}} | \varepsilon) = \begin{cases} 1 & \varepsilon \leq \frac{a}{2} \\ (0,1) & \frac{a}{2} < \varepsilon < \frac{a}{\sqrt{3}} \\ 0 & \varepsilon \geq \frac{a}{\sqrt{3}} \end{cases} \quad (4)$$

Alternatively, when both fcc and hcp sites are considered, the probability follows

$$P_{\text{fcc\&hcp}}(\mathbf{x}_{\text{adatom}} = \mathbf{x}_{\text{nearest}} | \varepsilon) = \begin{cases} 1 & \varepsilon \leq \frac{a}{2\sqrt{3}} \\ (0,1) & \frac{a}{2\sqrt{3}} < \varepsilon < \frac{a}{\sqrt{3}} \\ 0 & \varepsilon \geq \frac{a}{\sqrt{3}} \end{cases} \quad (5)$$

### Assignment and path planning method

Here we use existing python libraries for the Hungarian algorithm and the rapidly-exploring random tree (RRT) search algorithm to plan the manipulation path. For the Hungarian algorithm used for assigning each adatom to a target position, we use the linear sum assignment function in SciPy https://docs.scipy.org/doc/scipy-0.18.1/reference/generated/scipy.optimize.linear_sum_assignment.html. The cost matrix input for the linear sum assignment function is the Euclidean distance between each pair of adatom and target positions. Because the DRL agent is trained to manipulate atoms to target positions in any direction, we need to combine it with an any-angle path planning algorithm. We use the rapidly-exploring random tree (RRT) search algorithm implemented in the PythonRobotics python library https://github.com/AtsushiSakai/PythonRobotics/tree/master/PathPlanning. The RRT algorithm searches for paths between the adatom position and target position without colliding with other adatoms. However, it is worth noting that the RRT algorithm might not find optimal or near-optimal paths.

### Actions of trained agent

Here we analyze the mean and stochastic actions output by the trained DRL agent at the end of the training shown in Fig. 2a, b for 1000 states as shown in Suppl. Fig. 5. The target positions ($x_{\text{target}}$, $y_{\text{target}}$) are randomly sampled from the range used in the training and the adatom

**Table 1 | SAC hyperparameters**

| Parameter | Value |
|---|---|
| Optimizer | Adam[58] |
| Learning rate | $3 \times 10^{-4}$ |
| Number of hidden layers | 2 |
| Number of hidden units per layer | 256 |
| Number of sample per minibatch | 64 |
| Nonlinearity | ReLU |
| Replay buffer size | $10^6$ |
| Discount ($\gamma$) | 0.9 |
| Target smoothing coefficient ($\tau$) | 0.005 |

positions are set as $(x_{adatom}, y_{adatom}) = (0, 0)$. Several trends can be observed in the action variables output by the trained DRL agent. First, the agent intuitively favors using higher bias and conductance. During the training shown in Fig. 2, the DRL agent is observed to use increasingly large bias and conductance as shown in Suppl. Fig. 5. Also, analysis of the average bias and conductance over 100 episodes as functions of the number of episodes (see Suppl. Fig. 6) shows that the agent uses larger biases and conductance with increasing training episodes. Second, like in baseline manipulation parameters, the agent also moves the tip slightly further than the target position. But, different from the baseline tip movements (the tip moves to the target position extended by a constant length = 0.1 nm), the DRL agent moves the tip to the target position extended by a span that scales with the distance between the origin and the target. Fitting $x_{end}$ ($y_{end}$) as a function of $x_{target}$ ($y_{target}$) with a linear model yields $x_{end} = 1.02 x_{target} + 0.08$ and $y_{end} = 1.04 y_{target} + 0.03$ (indicated by the black lines in Suppl. Fig. 5b, c). Third, the agent also learns the variance each action variable can have while maximizing the reward. Finally, $x_{start}$, $y_{start}$, conductance, and bias show weak dependence on $x_{target}$ and $y_{target}$, which are however more difficult to interpret.

### Tip changes
During training, significant tip changes occurred due to the tip crashing deeply into the substrate surface and requiring tip apex reshape to perform manipulation using baseline parameters. It led to an abrupt decrease in the DRL agent's performance (shown in Fig. 2a, b) and changes in the tip height and topographic contrast in the STM scan (shown in Suppl. Fig. 7). After continued training, the DRL agent learns to adapt to the new tip conditions by manipulating with slightly different parameters as shown in Suppl. Fig. 8.

### Kagome lattice assembly
We built the kagome lattice in Fig. 3b by repeatedly building 8-atom units shown in Suppl. Fig. 9. In all, 8–15 manipulations were performed to build each unit, depending on the initial positions of the adatoms, the optimality of the path planning algorithm, and the performance of the DRL agent. Overall, 66 manipulations were performed to build the 42-atom kagome lattice with atomic precision. One manipulation together with the required STM scan takes roughly one minute. Therefore, the construction of the 42-atom kagome lattice takes around an hour, excluding the deposition of the Ag adatoms. The building time can be reduced by selecting a more efficient path planning algorithm and reducing STM scan time.

### Alternative reward design
In the training presented in the main text, we used a reward function (Eq. (1)) that is solely dependent on the manipulation error $\varepsilon = \|\mathbf{x}_{adatom} - \mathbf{x}_{target}\|$. During the experiment, we considered including a term $r' \propto (\mathbf{x}_{adatom,t+1} - \mathbf{x}_{adatom,t}) \cdot \mathbf{x}_{target}$ to the reward function to encourage the DRL agent to move the adatom toward the direction of the target. However, this term rewards the agent for moving the

adatom in the direction of the target even as it overshoots the target. When the $r'$ term is included in the reward function, the DRL agent trained for 2000 episodes shows a tendency to move the adatom overly far in the target direction as shown in Suppl. Fig. 10.

### Soft actor-critic
We implement the soft actor-critic algorithm with hyperparameters based on the original implementation[43] with small changes as shown in Table 1.

### Emphasizing recent experience replay
In the training the gradient descent update is performed in the end of each episode. We perform $K$ updates with $K$ = episode length. For update step $k = 0 \dots K\text{-}1$, we uniformly sample from the most recent $c_k$ data points according to the emphasizing recent experience replay sampling technique[45], where

$$c_k = \max(N \cdot \eta^{k \cdot \frac{1000}{K}}, c_{min}) \qquad (6)$$

where $N$ is the length of the replay buffer and $\eta$ and $c_{min}$ are hyperparameters used to tune how much we emphasize recent experiences set to 0.994 and 500, respectively.

### Hindsight experience replay
We use the 'future' strategy to sample up to three goals for replay[44]. For a transition $(s_t, a_t, r_t, s_{t+1})$ sampled from the replay buffer, max( episode length $- t$,3) goals will be sampled depending on the number of future steps in the episode. For each sampled goal, a new transition $(s'_t, a_t, r'_t, s'_{t+1})$ is added to the minibatch and used to estimate the gradient descent update of the critic and actor neural network in the SAC algorithm.

## Data availability
Data collected by and used for training the DRL agent, parameters of the trained neural networks, and codes to access them are available at https://github.com/SINGROUP/Atom_manipulation_with_RL.

## Code availability
The Python code package used to control the software, train the DRL agent and perform the automatic atom assembly is provided at https://github.com/SINGROUP/Atom_manipulation_with_RL.

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

## Acknowledgements

We thank Ondřej Krejčí, Jose L. Lado, and Robert Drost for fruitful discussions. The authors acknowledge funding from the Academy of Finland (Academy professor funding nos. 318995 and 320555) and the European Research Council (ERC-2017-AdG no. 788185 "Artificial Designer Materials"). This research was part of the Finnish Center for Artificial Intelligence FCAI. ASF has been supported by the World Premier International Research Center Initiative (WPI), MEXT, Japan. This research made use of the Aalto Nanomicroscopy Center (Aalto NMC) facilities and Aalto Research Software Engineering services.

## Author contributions

I.J.C. developed the software. M.A., A.K., and I.J.C. conducted the STM experiments and tested the code. I.J.C. and M.A. prepared the manuscript with input from A.K., A.I., P.L., and A.S.F.

## Competing interests

The authors declare no competing interests.
