## [Peer Review File · Nature Communications]

Response to Reviewers, first round -

Reviewer #1 (Remarks to the Author):

The authors have developed a reinforcement learning agent that can learn the lateral manipulation of adatoms to desired target positions. Such adatom arrangements of increasing complexity have been the basis of many fundamental experiments and findings in the past, and a further development of this technique could be interesting for fields like quantum information. The autonomous arrangement of such patterns potentially allows to push their size and complexity to a new level. Successful applications of machine learning and specifically reinforcement learning in scanning probe microscopy are yet scarce, but, in my opinion, have the potential to transform the field substantially. The manuscript is clearly written and exhibits a good scholarly presentation. Particularly promising is the ability of the agent to adapt to tip changes, a problem which is common in SPM experiments.

I can recommend publication of the work in Nature Communications. Nevertheless, I would like to emphasize two minor aspects that should be addressed prior to publication.

(1)

The idea to speed up the learning by avoiding time consuming STM scans in cases where the manipulation attempt has likely failed completely (i.e. where the adatom has not moved) is mentioned in the methods section and described in more detail in the S.I. However, in the main text this approach is only indicated by the single word "can", which can be overlooked very easily. The authors write on page 4 "Upon executing the action in the STM, a scan can be taken (see M. and S.I.) to update the adatom position [..]" I think that a sentence in the main text should be devoted to explaining the basic idea and the motivation behind the approach, leaving the details in Methods and S.I. What is the experience gained from such a failed action? Is $s_{t+1} = s_t$ assumed when there is no scan?

(2)

Could the authors learn anything from the strategies of the trained agent? In famous RL applications (Go, Starcraft, etc.), agents came up with strategies that were new to humans. In their paper the authors also use manually tuned baseline parameters for comparison and, so far, everyone else in the community does the same. Therefore, I think it would be interesting to know if the agent came up with some "innovative" strategy with respect to any of the parameters G , V , x_{start} , and x_{end} such that the machine learning could also be complemented by human learning.

One additional point:

I understand that the manipulation is considered a success if $e < 0.166$ nm, but I do not understand the sentence on pages 4/6, which reads "As a result, the atom manipulation error, [..], is limited from 0 nm to 0.166 nm, [..]"

What do the authors mean if they say that the error (distance between two positions) is "limited"?

Reviewer #2 (Remarks to the Author):

This is an impressive paper -- the authors' approach to reinforcement learning-enabled atomic manipulation is elegant and compelling. I am confident that their work will have a major impact in both the SPM and, more broadly, nanoscience fields.

I enthusiastically recommend publication. I have just a few minor points to raise:

(i) Second paragraph in Main: "...for each new adatom/surface and tip apex combination."

(ii) p.3, first paragraph. I'd quibble with regard to Ref. 35 being described as not showing "performances that are optimal or exceed conventional algorithmic or manual approaches."

Although the approach adopted in Ref. 35 perhaps does not have the same level of autonomy as described in the authors' manuscript, I'd argue that it exceeds conventional algorithmic or manual approaches in that the correct (and entirely non-trivial) tip trajectory was indeed learned. This, however, is a somewhat pedantic point.

(iii) p.4, line 5: Why were five manipulations chosen? Why five in particular?

(iv) nc-AFM provides much richer information on tip-sample interactions than that achievable via STM. Have the authors attempted to use nc-AFM channels (frequency shift, amplitude/dissipation) in their RL algorithms?

(v) p.6, line 7: The reward term is, of course, absolutely key to the entire RL process. Did the authors experiment with other reward strategies? If so, it would be helpful for other researchers if a brief description could be added to the SI as regards the relative efficacy of other approaches.

(vi) p. 8, caption to Fig. 2, line 3: What is meant by "significant" tip change in this context?

(vii) p. 9, first paragraph. Is it possible for the system to learn about the manipulation capability of different tip states? For example, could the system rule out a particular tip apex right from the start so as not to waste time with a sub-optimal probe structure?

(ix) p.9, Fig. 3. As a Douglas Adams fan, I have to ask this: Was the choice of a 42-atom structure entirely coincidental...?

(x) p.9, Fig. 3(b). It seems from the contrast variations that a couple of sites on the lattice have Ag dimers rather than single atoms. (There appear to be slightly brighter maxima in the bottom left and top right). Is this correct? If so (or even if not!), how does the algorithm deal with unwanted events like this, where one adatom gets too close to another adatom during manipulation and irreversibly forms a dimer?

There are all, however, minor points. The authors are to be congratulated on a beautiful piece of work.

Reviewer #3 (Remarks to the Author):

This paper describes the use of deep reinforcement learning (DRL) for automated assembly of adatoms of Ag on a surface of Ag. In terms of test systems for DRL for autonomous fabrication at the atomic scale, this is probably one of the simpler systems (and rightfully so, given the nascent nature of DRL in the field of materials science more generally).

The results are well presented, showing that DRL methods appear to outperform a baseline method where the parameters are simply chosen beforehand. However, I must question whether this baseline is appropriate, and whether DRL is the right choice here to tackle this problem. The authors note that it takes about 6000 manipulations to train the agent. If instead, there were a limited number of manipulations after each tip change and a standard optimization algorithm was used to find the best parameters for that particular tip, would this not be more efficient?

I guess my struggle is in understanding what state information is present that enables tuning of the parameters for reliable manipulation. Without such information present in the state, it seems doubtful that the DRL agent is robust - rather, it adapts once the tip has changed, but it is not likely to revert to strong performance if the tip changed back to some earlier, original state without more training. Maybe I am missing something, but I would have thought that the state should incorporate information about the local neighborhood and current traces so that a mapping between the state and actions could be found that is robust against tip changes. Perhaps the authors believe this information is only encoded in the state transitions themselves. I would tend to disagree, it should also manifest in subtle changes in the imaging conditions too.

Therefore, I have a request for an additional experiment: can the authors continue to test the agent (without training) until the tip changes at least two or three more times? If the agent is still performing well, then the policy is robust. I am confused in Figure 2d,e because the text states that this is the result of the trained agent, but then references 'retraining', so I could not understand what was being communicated.

Overall I am very satisfied with this manuscript, but believe that without a proper estimate of the robustness, it is not yet suitable for publication. Finally I commend the authors on publishing the code for the paper. We need more such people in our community.

- RKV

Reviewer #4 (Remarks to the Author):

The manuscript "Precise atom manipulation through deep reinforcement learning" by I-Ju Chen et al. focuses on autonomous atom manipulation in a scanning tunneling microscope.

Atom manipulation is an essential tool to build and study atomically-precise structures and devices. The technique will only become more important the next years now that it has been established for both condensed-matter research (e.g. band structure engineering) and quantum information science (single-atom qubits). STM is the single available technique suitable for manipulating adatoms and molecules on surfaces with atomic precision. However, the procedure is tedious: 1) a sufficiently stable tip needs to be prepared, expected to be suitable for manipulation, 2) the atom manipulation parameters need to be established, and 3) the atomic structure itself needs to be assembled.

This manuscript automates 2) and 3): finding the manipulation parameters and building the structure using a combination of reinforcement learning techniques and path-planning algorithms. Out of these two, 2) is new and presents the most important part of the paper. Path-planning algorithms for the assembly 3) have been used independently before.

The manuscript demonstrates the success of the automated manipulation for Ag on Ag(111) using 3 different tips, and for Co on Ag(111).

It will be useful to indicate how large the parameter change was to achieve the end result, and to display (Sup Mat) how drastic the occurring tip changes were. Further comments can be found below.

Recommendation:

The results with a high success rate are convincing and help enable automating labor-intensive processes for several research groups. However, the paper mainly presents a useful and quite specific tool. The further scientific content itself is thin. I thus recommend the paper to be published in another journal, such as RSI.

Comments and suggestions:

- P. 2, line 12: Ref. 11: as far as I am aware, this paper is not about atom manipulation. It is notoriously hard to achieve well-defined manipulation on graphene without picking up the atoms and without perturbations while scanning, and it has been successfully demonstrated only for a few types of adatoms, such as H (DOI: 10.1126/science.aad8038) and Co (doi: 10.1021/acsnano.6b05823).

- P. 2, line 12: Optionally, semiconductor surfaces could be mentioned (Stefan Fölsch, etc.)

- P. 2, line 15: Another strong application of 'atom manipulation' is STM hydrogen lithography for qubit devices (Michelle Simons, Robert Wolkow, etc.)
- P. 5, Fig. 1c: The full height of the Ag adatom is not clear, as the color scale seems clipped at 68.8 pm (the actual measured height exceeds this value, displayed in white).
- P. 6, line 5-6: Shouldn't the bottom option be "<" instead of "<="? The main text also indicates 'smaller' (not smaller or equal too), which makes sense as otherwise a neighboring hollow site would have been reached.
- P. 8, Fig. 2: Interesting and useful information to assess the method would be:
 - 1) Fig. 2a: Indicate how much the manipulation parameters changed overall (and especially upon tip changes)
 - 2) Sup Mat: provide the images taken right after the tip changes. My impression is that the tip changes were not too drastic?
- P. 9, Fig. 3:
 - 1) Fig. 3a: It would be useful to add a linecut (radially integrated?), showing a high peak in the center, low neighboring peaks at $a/\sqrt{3}$, intermediate peaks at a , and lowest peaks at $2a/\sqrt{3}$.
 - 2) Fig. 3b: Color bar is missing.
 - 3) Optional: it would be interesting to add a schematic, showing the underlying atomic lattice (not chosen optimally) and the target positions and actual positions. The lattice rotation would reflect the orientation of Fig. 3a.
 - 4) Optional: it could be interesting to see a scan with the original positions of the adatoms and an indication of their manipulation paths before reaching Fig. 3b (Sup Mat.).
- P. 10, line 14: "a robust and efficient technique": quantify 'how efficient' somewhere (Sup Mat)? (E.g. how long did it take to prepare the kagome lattice in Fig. 3b from scratch?)
- P. 11, line 11: add "for the baseline data".
- Sup. Mat., P. 1: "high-bias range": please add for the reader why this range was limited to a lower tunneling conductance (e.g. the preamp range, avoiding tip changes, avoiding heating, etc.).
- Sup. Mat., P. 4: "integrate"  "integrating"
- Sup. Mat., Fig. S4, bottom left:

useful to indicate " $a/2$ " with a circle as well.

REVIEWER #1:

Overall comments: The authors have developed a reinforcement learning agent that can learn the lateral manipulation of adatoms to desired target positions. Such adatom arrangements of increasing complexity have been the basis of many fundamental experiments and findings in the past, and a further development of this technique could be interesting for fields like quantum information. The autonomous arrangement of such patterns potentially allows to push their size and complexity to a new level. Successful applications of machine learning and specifically reinforcement learning in scanning probe microscopy are yet scarce, but, in my opinion, have the potential to transform the field substantially. The manuscript is clearly written and exhibits a good scholarly presentation. Particularly promising is the ability of the agent to adapt to tip changes, a problem which is common in SPM experiments.

I can recommend publication of the work in Nature Communications. Nevertheless, I would like to emphasize two minor aspects that should be addressed prior to publication.

Author reply: We thank the reviewer for the positive assessment and constructive criticism of our work. Please find our point-by-point responses to the reviewer’s comments below.

Reviewer comment 1. The idea to speed up the learning by avoiding time consuming STM scans in cases where the manipulation attempt has likely failed completely (i.e. where the adatom has not moved) is mentioned in the methods section and described in more detail in the S.I. However, in the main text this approach is only indicated by the single word “can”, which can be overlooked very easily. The authors write on page 4 “Upon executing the action in the STM, a scan can be taken (see M. and S.I.) to update the adatom position [..].“ I think that a sentence in the main text should be devoted to explaining the basic idea and the motivation behind the approach, leaving the details in Methods and S.I. What is the experience gained from such a failed action? Is $s_{(t+1)} = s_t$ assumed when there is no scan?

Author reply: We thank the reviewer for pointing out this issue. We have elaborated more on using a classification method to determine whether a scan should be taken in the revised manuscript. In case a scan is not taken, the state is considered unchanged with $s_{t+1} = s_t$. The following sentences are modified/ added on P.4.

Old text: Upon executing the action in the STM, a scan can be taken (see Methods and Supplementary Information) to update the adatom position to form the new state s_{t+1} and the agent receives a reward $r_t(s_t, a_t, s_{t+1})$.

New text: Upon executing the action in the STM, a method combining a convolutional neural network and an empirical formula is used to classify whether the adatom has likely moved from the tunneling current measured during manipulation (see Methods and Supplementary Information). If the method determines the adatom has likely moved, a scan is taken to

update the adatom position to form the new state s_{t+1} . Otherwise, the scan is often skipped to save time and the state is considered unchanged $s_{t+1} = s_t$. The agent then receives a reward $r_t(s_t, a_t, s_{t+1})$.

Reviewer comment 2. Could the authors learn anything from the strategies of the trained agent? In famous RL applications (Go, Starcraft, etc.), agents came up with strategies that were new to humans. In their paper the authors also use manually tuned baseline parameters for comparison and, so far, everyone else in the community does the same. Therefore, I think it would be interesting to know if the agent came up with some “innovative” strategy with respect to any of the parameters G , V , x_start , and x_end such that the machine learning could also be complemented by human learning.

Author reply: We agree with the reviewer that it is indeed interesting to analyze the strategies learned by the agent. We now include an analysis of the trained DRL agent’s actions as functions of \mathbf{x}_{target} in the **Actions of trained agent** section in the SI. We observe

- The agent intuitively favors using higher bias and conductance.
- The agent also moves the tip slightly further than the target position. But, different from the baseline (the tip moves to the target position extended by a constant length = 0.1 nm), the DRL agent moves the tip to the target position extended by a span that scales with the distance between the original adatom positions and the target.
- The agent learns the variance each action variable can have while maximizing the reward.
- x_{start} , y_{start} , conductance, and bias show weak dependence on x_{target} and y_{target} . But they are more difficult to interpret.

”Additional point” from the Reviewer. I understand that the manipulation is considered a success if $e < 0.166$ nm, but I do not understand the sentence on pages 4/6, which reads “As a result, the atom manipulation error, [..], is limited from 0 nm to 0.166 nm, [..]” What do the authors mean if they say that the error (distance between two positions) is “limited”?

Author reply: We thank the reviewer for pointing out this unclear phrasing. We meant to say that depending on the target position relative to the underlying adsorption site position, the achievable atom manipulation error is limited from 0 nm to 0.166 nm. To clarify this, we modify the following sentence on P. 6:

Old text: As a result, the atom manipulation error ε ,

New text: As a result, the optimal atom manipulation error ε ,

REVIEWER #2:

Overall comments: This is an impressive paper – the authors’ approach to reinforcement learning-enabled atomic manipulation is elegant and compelling. I am confident that their work will have a major impact in both the SPM and, more broadly, nanoscience fields.

I enthusiastically recommend publication.

Author reply: We thank the reviewer for their positive outlook on our work and its potential impact. Please find our point-by-point responses to the Reviewers comments below.

Reviewer comment (i). Second paragraph in Main: "...for each new adatom/surface and tip apex combination."

Author reply: We thank the reviewer for pointing out this detail: the tip apex is indeed critical for successful atom manipulation. We have now modified the sentence according to the reviewer’s suggestion on P. 2.

Reviewer comment (ii). p.3, first paragraph. I’d quibble with regard to Ref. 35 being described as not showing "performances that are optimal or exceed conventional algorithmic or manual approaches." Although the approach adopted in Ref. 35 perhaps does not have the same level of autonomy as described in the authors’ manuscript, I’d argue that it exceeds conventional algorithmic or manual approaches in that the correct (and entirely non-trivial) tip trajectory was indeed learned. This, however, is a somewhat pedantic point.

Author reply: We thank the reviewer for pointing out this misleading phrasing. We agree with the reviewer that the RL agent in ref. 38 has successfully learned the non-trivial tip trajectory required to achieve the molecule removal task. We only describe the RL agents’ performance in ref. 37 and 38 as not exceeding conventional algorithmic or manual approaches because no baseline performances were defined, thus it’s unclear if the RL agents achieved better performances.

We have removed the description on P. 3 in the main text to avoid confusion.

Reviewer comment (iii). p.4, line 5: Why were five manipulations chosen? Why five in particular?

Author reply: The episode length is a hyperparameter that should ideally be tuned to minimize the overall training time. However, hyperparameter tuning is very challenging in time-consuming real-world training. Therefore, here the episode length is set to five based on the intuition that the agent should have several attempts to accomplish the goal, but the episode length shouldn’t be so long that it gets stuck in trying to solve an overly challenging

episode. The following texts are modified on P. 4 in the main text to explain this design decision:

Old text: ... the agent can perform up to five manipulations to accomplish the task.

New text: ... the agent can perform up to N manipulations to accomplish the task. Here the episode length is set to an intermediate value $N = 5$ that allows the agent to attempt different ways to accomplish the goal without it being stuck in overly challenging episodes.

Reviewer comment (iv). nc-AFM provides much richer information on tip-sample interactions than that achievable via STM. Have the authors attempted to use nc-AFM channels (frequency shift, amplitude/dissipation) in their RL algorithms?

Author reply: We agree with the reviewer that nc-AFM channels provide richer information on tip-sample interactions. nc-AFM is also applicable to insulating substrates, which are interesting avenues for atom manipulation experiments. However, including more state variables will also increase the dimensionality and thus the complexity of the RL problem, resulting in slower training. It is an interesting open question that would require a dedicated study and something we are planning in the future, but falls outside the scope of this manuscript.

Reviewer comment (v). p.6, line 7: The reward term is, of course, absolutely key to the entire RL process. Did the authors experiment with other reward strategies? If so, it would be helpful for other researchers if a brief description could be added to the SI as regards the relative efficacy of other approaches.

Author reply: We agree with the reviewer that the reward term is key to the RL training process. We experimented with an alternative reward design that includes a term $r' \propto (\mathbf{x}_{adatom,t+1} - \mathbf{x}_{adatom,t}) \cdot \mathbf{x}_{target}$ to the reward function to encourage the DRL agent to move the adatom toward the direction of the target. However, this term rewards the agent for moving the adatom in the direction of the target even as it overshoots the target. As a result, the trained DRL agent exhibits sub-optimal performance and moves the atom overly far in the target direction.

We now include a description of this alternative reward design and its effect on the training result in the **Alternative reward design** section in the SI.

Reviewer comment (vi). p. 8, caption to Fig. 2, line 3: What is meant by "significant" tip change in this context?

Author reply: We thank the reviewer for pointing out this unclear description. Here we refer to significant tip change as when the tip is crashed deeply into the substrate either intentionally or by accident, making manipulation not possible with the baseline parameters,

FIG. R1. The STM scans taken before and after tip changes led to tip conditions named tip 1, tip 2, and tip 3 as discussed in Fig. 2(a, b, d, e). Changes in the tip height and topographic contrast (e.g. double-tip in the first scan) can be observed.

and the tip has to be reshaped by an operator. The changes can be observed in the difference in the scan (see Fig. R1) and now included in the **Tip changes** section in the SI.

The following text in the caption of Fig. 2 (a,b) on P. 9 is modified to clarify the meaning of 'significant' tip change.

Old text: The arrows indicate significant tip changes during training.

New text: The arrows indicate significant tip changes which occurred when the tip crashed deeply into the substrate and the tip apex needed to be reshaped to perform manipulation with the baseline parameters (see Methods). The changes can be observed in the STM scans (see SI).

Reviewer comment (vii). p. 9, first paragraph. Is it possible for the system to learn about the manipulation capability of different tip states? For example, could the system rule out a particular tip apex right from the start so as not to waste time with a sub-optimal probe structure?

Author reply: Being able to assess the manipulation capability of different tip states would be a valuable feature for our system. We believe this can be achieved by including the tunneling current traces and/or the STM images as input and using either a modified DRL algorithm or a separate algorithm that can be based on neural networks similar to Alldritt, B., et. al, Automated tip functionalization via machine learning in scanning probe microscopy, Comput. Phys. Commun. 2022, 273, 108258 and Krull, A., et. al, Artificial-intelligence-driven scanning probe microscopy. Communications Physics 2020, 3, 54.

Reviewer comment (ix). p.9, Fig. 3. As a Douglas Adams fan, I have to ask this: Was the choice of a 42-atom structure entirely coincidental...?

Author reply: We refuse to answer that question on the grounds that we don't know the answer.

Reviewer comment (x). p.9, Fig. 3(b). It seems from the contrast variations that a couple of sites on the lattice have Ag dimers rather than single atoms. (There appear to be slightly brighter maxima in the bottom left and top right). Is this correct? If so (or even if not!), how does the algorithm deal with unwanted events like this, where one adatom gets too close to another adatom during manipulation and irreversibly forms a dimer?

Author reply: We thank the reviewer for this question. The path planning algorithm – rapidly exploring random tree search - is responsible for planning a manipulation path that prevents the atoms from colliding with other atoms (i.e. distance between the two atoms $> 2 \times$ atom neighborhood radius). The probability of accidental collision decreases with increasing atom neighborhood radius. In our experiment, the atom neighborhood radius is empirically set as 0.5 nm.

We agree with the reviewer that a few sites in Fig. 3(b) appear to be dimers. We believe these dimers are formed during deposition on the surface by controlled tip indentations (Limot et al, PRL (2005), 94, 126102). Even with these imperfections, the structure presented in Fig. 3(b) serves as an example of our autonomous atom assembly system. To make this clearer we modified the text on P. 10 as follows.

Old text: The RRT algorithm plans the paths between the start and final positions of 5 the adatom while avoiding collisions between adatoms.

New text: The RRT algorithm plans the paths between the start and final positions of the adatom while avoiding collisions between adatoms - note that it is possible that the structure in Fig. 3(b) contains 1 or 2 dimers, but these were likely formed before the manipulation started as the agent avoids atomic collisions.

Other comments of the reviewer. There are all, however, minor points. The authors are to be congratulated on a beautiful piece of work.

Author reply: We thank the reviewer for their constructive criticism and positive assessment of our manuscript.

REVIEWER #3:

Overall comments: This paper describes the use of deep reinforcement learning (DRL) for automated assembly of adatoms of Ag on a surface of Ag. In terms of test systems for DRL for autonomous fabrication at the atomic scale, this is probably one of the simpler systems (and rightfully so, given the nascent nature of DRL in the field of materials science more generally).

Author reply: We thank the referee for supporting our choice of system.

Reviewer comment 1. The results are well presented, showing that DRL methods appear to outperform a baseline method where the parameters are simply chosen beforehand. However, I must question whether this baseline is appropriate, and whether DRL is the right choice here to tackle this problem. The authors note that it takes about 6000 manipulations to train the agent. If instead, there were a limited number of manipulations after each tip change and a standard optimization algorithm was used to find the best parameters for that particular tip, would this not be more efficient?

Author reply: Deep reinforcement learning and standard optimization methods differ fundamentally in that DRL is designed to achieve optimal control of sequential decision problems while standard optimization methods generally search for one single set of parameters that will result in the best outcome. We think it's suitable to formulate precise atom manipulation as a sequential decision problem since it often requires multiple consecutive manipulations to move atoms to their target positions precisely, particularly for more challenging tip-substrate combinations.

Additionally, by using deep neural networks, DRL will also have the ability to generalize from its experience and adapt to new tip configurations faster. Finally, it's worth noting that basic optimization methods such as grid search might also require large numbers of evaluations, on the order of $n^{D_{state}+D_{action}}$ with the number of grid points for each dimension n and the state/ action dimension D_{state}/ D_{action} , to find the best set of manipulation parameters.

Reviewer comment 2. I guess my struggle is in understanding what state information is present that enables tuning of the parameters for reliable manipulation. Without such information present in the state, it seems doubtful that the DRL agent is robust - rather, it adapts once the tip has changed, but it is not likely to revert to strong performance if the tip changed back to some earlier, original state without more training. Maybe I am missing something, but I would have thought that the state should incorporate information about the local neighborhood and current traces so that a mapping between the state and actions could be found that is robust against tip changes. Perhaps the authors believe this information is only encoded in the state transitions themselves. I would tend to disagree, it should also manifest in subtle changes in the imaging conditions too.

Therefore, I have a request for an additional experiment: can the authors continue to test the agent (without training) until the tip changes at least two or three more times? If the agent is still performing well, then the policy is robust.

Author reply: We thank the reviewer for giving us the opportunity to clarify our statement about the robustness of the DRL agent. We refer to the DRL agent’s robustness as both its relatively good and near-optimal performance before and after sufficient continued training (Fig. 2(d,e)). The DRL agent’s ability to continually train under new environment conditions is one of the key reasons why the DRL agent is robust against tip changes.

We modify the following sentence on P.8 to clarify this point:

Old text: The results show that the RL algorithm is more robust and adaptable against tip changes than fixed manipulation parameters.

New text: The results show that, with continued training, the RL algorithm is more robust and adaptable against tip changes than fixed manipulation parameters.

We agree with the reviewer that the DRL agent can be made more robust against tip changes if more information about tip-substrate interaction is included in the state variable. However, directly including the current trace and/or STM scans will drastically increase the state dimension and consequently the amount of training data needed. More experiments beyond the scope of this work are needed to encode the tip-substrate information efficiently.

In this work, the agent indeed only learns the tip-substrate interaction through state transitions. Therefore, continued training is essential for the agent to maintain good performance and the suggested experiment would not be an accurate demonstration of reliability. Besides, several design choices were made so that the DRL agent’s performance is stable and improves rapidly under new tip conditions. First, SAC agents are trained to succeed at their tasks while acting as randomly as possible, so they are able to generalize to perturbations in the environment better. Second, the RL agents are trained using all the experience in the replay buffer while prioritizing more recent experiences. Therefore, the agent is able to adapt to new tip conditions efficiently without forgetting previous tip conditions it encountered.

Reviewer comment 3. I am confused in Figure 2d,e because the text states that this is the result of the trained agent, but then references ‘retraining’, so I could not understand what was being communicated.

Author reply: We thank the reviewer for pointing out this unclear phrasing. In Fig. 2(d,e), we compared the performance of the baseline with the DRL agent, which is continually trained following three tip changes.

To clarify this point, the following texts are modified:

Fig. 2(d,e) caption

Old text: The trained RL agent and baseline are compared under three tip conditions. ... In contrast, RL always converges to near-optimal performances after sufficient training.

New text: The RL agent, which is continually trained, and the baseline are compared under three tip conditions that resulted from the tip changes indicated in (a, b). ... In contrast, RL always converges to near-optimal performances after sufficient continued training.

P. 8

Old text: In contrast, the RL agent maintains relatively good performance before re-training and reaches success rates $> 95\%$ after re-training.

New text: In contrast, the RL agent maintains relatively good performances within the first 100 episodes of continued training and eventually reaches success rates $> 95\%$ after more training under the new tip condition.

Other comments of the reviewer. Overall I am very satisfied with this manuscript, but believe that without a proper estimate of the robustness, it is not yet suitable for publication. Finally I commend the authors on publishing the code for the paper. We need more such people in our community.

Author reply: We thank the reviewer for their constructive comments, and hope that the revised version of the manuscript addresses any concerns about the robustness of our method. We hope that the freely accessible code will benefit the wider SPM community and enable atom manipulation on novel substrate/adsorbate/tip combinations.

REVIEWER #4:

Overall comments: The manuscript “Precise atom manipulation through deep reinforcement learning” by I-Ju Chen et al. focuses on autonomous atom manipulation in a scanning tunneling microscope.

Atom manipulation is an essential tool to build and study atomically-precise structures and devices. The technique will only become more important the next years now that it has been established for both condensed-matter research (e.g. band structure engineering) and quantum information science (single-atom qubits). STM is the single available technique suitable for manipulating adatoms and molecules on surfaces with atomic precision. However, the procedure is tedious: 1) a sufficiently stable tip needs to be prepared, expected to be suitable for manipulation, 2) the atom manipulation parameters need to be established, and 3) the atomic structure itself needs to be assembled. This manuscript automates 2) and 3): finding the manipulation parameters and building the structure using a combination of reinforcement learning techniques and path-planning algorithms. Out of these two, 2) is new and presents the most important part of the paper. Path-planning algorithms for the assembly 3) have been used independently before. The manuscript demonstrates the success of the automated manipulation for Ag on Ag(111) using 3 different tips, and for Co on Ag(111).

It will be useful to indicate how large the parameter change was to achieve the end result, and to display (Sup Mat) how drastic the occurring tip changes were. Further comments can be found below. [...]

The results with a high success rate are convincing and help enable automating labor-intensive processes for several research groups. However, the paper mainly presents a useful and quite specific tool. The further scientific content itself is thin. I thus recommend the paper to be published in another journal, such as RSI.

Author reply: We thank the reviewer for their constructive criticism and their summary of our work in relation to motivations and methods in atomically precise structure assembly. However, as supported by the three other referees, we strongly believe that the revised manuscript is worthy of publication in the wider context of Nature Communications. Please find our point-by-point responses to the reviewer’s comments below.

Reviewer comments, pages 2-6:

- P. 2, line 12: Ref. 11: as far as I am aware, this paper is not about atom manipulation. It is notoriously hard to achieve well-defined manipulation on graphene without picking up the atoms and without perturbations while scanning, and it has been successfully demonstrated only for a few types of adatoms, such as H (DOI: 10.1126/science.aad8038) and Co (doi: 10.1021/acsnano.6b05823).

- P. 2, line 12: Optionally, semiconductor surfaces could be mentioned (Stefan Fölsch, etc.)
- P. 2, line 15: Another strong application of ‘atom manipulation’ is STM hydrogen lithography for qubit devices (Michelle Simons, Robert Wolkow, etc.)
- P. 5, Fig. 1c: The full height of the Ag adatom is not clear, as the color scale seems clipped at 68.8 pm (the actual measured height exceeds this value, displayed in white).
- P. 6, line 5-6: Shouldn’t the bottom option be “<” instead of “<=”? The main text also indicates ‘smaller’ (not smaller or equal too), which makes sense as otherwise a neighboring hollow site would have been reached.

Author reply: We thank the reviewer for pointing out these literature references and inaccuracies. We have addressed these issues in the revised version of the manuscript.

Reviewer comments, page 8:

- P. 8, Fig. 2: Interesting and useful information to assess the method would be: 1) Fig. 2a: Indicate how much the manipulation parameters changed overall (and especially upon tip changes) 2) Sup Mat: provide the images taken right after the tip changes. My impression is that the tip changes were not too drastic?

Author reply: We now include the manipulation parameters after continued training and the STM scans under different tip conditions in the **Tip changes** section, Fig. S7, and Fig. S8 in the SI. Changes in the tip height and topographic contrast can be observed between the STM scans, indicating a change in the tip structure. Even though the manipulation parameters don’t change drastically, the evident performance improvement after continued training suggests that the slight changes significantly impact the manipulation precision.

Reviewer comments, page 9: - P. 9, Fig. 3:

- 1) Fig. 3a: It would be useful to add a linecut (radially integrated?), showing a high peak in the center, low neighboring peaks at $a/\sqrt{3}$, intermediate peaks at a , and lowest peaks at $2a/\sqrt{3}$.

Author reply: The linecuts from the center to the nearest and the second nearest neighbor sites are added in Fig. 3(a).

- 2) Fig. 3b: Color bar is missing.

Author reply: We have now included the color bar in the figure.

- 3) Optional: it would be interesting to add a schematic, showing the underlying atomic lattice (not chosen optimally) and the target positions and actual positions. The lattice rotation would reflect the orientation of Fig. 3a.

Author reply: Considering the scale of Fig. 3(b), the underlying atomic lattices will appear very dense and not clearly visible. Therefore, we opt not to show them together with the Kagome lattice. Instead, a point contact scan showing the underlying atomic lattice is added to Fig. 3. It shows a lattice orientation consistent with Fig. 3(a).

4) Optional: it could be interesting to see a scan with the original positions of the adatoms and an indication of their manipulation paths before reaching Fig. 3b (Sup Mat.).

Author reply: We now include the STM images of the original and final positions of the adatoms that form part of the Kagome lattice in Fig. S8 in the SI. Manipulation paths taken by the DRL agent are indicated in the image.

Reviewer comments, pages 10-11 + Supplementary: - P. 10, line 14: “a robust and efficient technique”: quantify ‘how efficient’ somewhere (Sup Mat)? (E.g. how long did it take to prepare the kagome lattice in Fig. 3b from scratch?)

Author reply: The DRL agent performed 66 manipulations, which took around an hour, to build the 42-atom Kagome lattice with atomic precision as shown in Fig. 3(b). We now include a more detailed description of the building process of the 42-atom Kagome lattice and directions to improve the efficiency in section **Kagome lattice assembly** in the SI.

- P. 11, line 11: add “for the baseline data”.

Author reply: We have adjusted the text as suggested by the reviewer. To clarify, the tip conditioning described in Methods is performed prior to both the baseline manipulation tests and the RL training sequences.

- Sup. Mat., P. 1: “high-bias range”: please add for the reader why this range was limited to a lower tunneling conductance (e.g. the preamp range, avoiding tip changes, avoiding heating, etc.).

Author reply: The manipulation of cobalt atoms can be performed in the “conventional” low bias (\sim mV) and high conductance range ($\sim \mu\text{A}/\text{V}$), and alternatively with a high bias and lower conductance setpoint. As shown in Limot & Berndt, Appl. Surf. Sci. vol 237, p. 572-576, 2004, tunneling conductance $\sim \text{nA}/\text{V}$ is sufficient to manipulate Co atoms on Ag(111) with a high bias ($\sim \text{V}$) due to a different manipulation mechanism. Therefore, we chose a lower conductance range in combination with a high bias range, which is sufficient for manipulating Co atoms while lowering the risk of tip changes and crashes to the surface.

To clarify this point, we add the following discussion on P.2 of the SI:

In the high-bias regime, a significantly lower tunneling conductance is sufficient to manipulate Co atoms due to the different manipulation mechanism. In addition, a high bias ($\sim \text{V}$) combined with a higher tunneling conductance ($\sim \mu\text{A}/\text{V}$) might lead to tip and substrate damage.

- Sup. Mat., P. 4: “integrate” → “integrating”

Author reply: The grammar mistake has been fixed.

- Sup. Mat., Fig. S4, bottom left: useful to indicate “ $a/2$ ” with a circle as well.

Author reply: A circle with $a/2$ radius is now added in Fig. S4.

Response to Reviewers, second round -

Reviewer #1 (Remarks to the Author):

The authors have addressed all referee comments and suggestions in detail, particularly extending the SI, to which four new sections have been added. Given the overall positive referee comments, I think that the updated paper can be published in its present form.

Reviewer #2 (Remarks to the Author):

The authors have convincingly and comprehensively addressed all reviewers' comments. I have no hesitation in recommending the paper for publication.

Reviewer #4 (Remarks to the Author):

The authors have adequately addressed most points raised in my review. The well-written manuscript has further improved in quality and comprehensiveness.

I recommend publication in Nature Communications.

For the final version of the manuscript, I request the authors to incorporate the following:

1. The authors included the DRL actions after continued training (Fig. S7). Since the manuscript focuses on the improvement of the manipulation performance from the baseline, it is additionally relevant to show the average bias change and average conductance change (trend) from the baseline values with increasing number of episodes, which one can relate to Fig. 2a, including for the drastic improvement the first 1000 episodes. (See my initial comment 'how large was the parameter change to achieve the end result?')

2. In the chosen experimental approach, the target positions are uniformly distributed, defined 'blindly' without prior notion of the positions of the adsorption sites. This means that the following factors contribute to the manipulation error: (a) a deviation of the target position from the fcc/hcp adsorption sites and (b) manipulation imprecisions related to suboptimal manipulation parameters for the given tip apex / adatom / substrate combination. The DRL agent can be expected to learn to: (a) adjust the tip end position to the adsorption sites, leading to more reliable manipulations (more efficiently avoiding snapping onto an fcc/hcp site just beyond $a/\sqrt{3}$); and (b) optimize all manipulation parameters for the given system. In the final manuscript, could the authors comment on the role of "(a)" for the improved performance (unrelated to the tip/adatom/substrate interaction), especially in the first 1000 episodes?

Now that the manuscript will likely reach its final state soon, I would like to point out a few last typos: P.4, 21: r_t, s_{t+1} . P. 7, 22: a given $\{a\}$ manipulation error.

I support and appreciate the open-source availability of the code package.

REVIEWER #4:

Overall comments: The authors have adequately addressed most points raised in my review. The well-written manuscript has further improved in quality and comprehensiveness.

I recommend publication in Nature Communications. For the final version of the manuscript, I request the authors to incorporate the following:

Reviewer comment 1. The authors included the DRL actions after continued training (Fig. S7). Since the manuscript focuses on the improvement of the manipulation performance from the baseline, it is additionally relevant to show the average bias change and average conductance change (trend) from the baseline values with increasing number of episodes, which one can relate to Fig. 2a, including for the drastic improvement the first 1000 episodes. (See my initial comment ‘how large was the parameter change to achieve the end result?’)

Author reply: We thank the reviewer for the suggestion. We have now included the average bias and conductance over 100 episodes as functions of the number of episodes shown below in Fig. S6 (section **Actions of trained agent**). The figure shows that the agent uses larger biases and conductance with increasing training episodes.

FIG. R1. **Average bias and conductance used by DRL agent** The rolling mean (over 100 episodes) of the biases and conductance used by the DRL agent in the training shown in Fig. 2. As the training progresses, the DRL agent applies larger bias and conductance to manipulate the atom.

Reviewer comment 2. In the chosen experimental approach, the target positions are uniformly distributed, defined ‘blindly’ without prior notion of the positions of the adsorption sites. This means that the following factors contribute to the manipulation error: (a) a deviation of the target position from the fcc/hcp adsorption sites and (b) manipulation imprecisions related to suboptimal manipulation parameters for the given tip apex / adatom / substrate combination. The DRL agent can be expected to learn to: (a) adjust the tip end position to the adsorption sites, leading to more reliable manipulations (more efficiently avoiding snapping onto an fcc/hcp site just beyond $a/\sqrt{3}$); and (b) optimize all manipulation parameters for the given system. In the final manuscript, could the authors comment on the role of “(a)” for the improved performance (unrelated to the tip/adatom/substrate interaction), especially in the first 1000 episodes?

Author reply: We agree with the reviewer that the DRL agent can be expected to learn to adjust the tip end position to move the atom to the adsorption sites. However, the evidence is elusive in the collected training data. When visualizing the tip end position (relative to the atom position) distribution in Gaussian kernel density estimation plots as shown below, there is no clear correspondence with the adsorption site positions shown in Fig. 3a. As we cannot conclude whether the agent has learned to move the atoms to the adsorption site, we do not include this discussion in the manuscript for better readability.

Reviewer comment 3. Now that the manuscript will likely reach its final state soon, I would like to point out a few last typos: P.4, 21: r_t, s_{t+1} . P. 7, 22: a given $\{a\}$ manipulation error.

Author reply:

The typos pointed out by the reviewer have been corrected in the final version of the manuscript.